# Beyond BMI

**DOI:** 10.3390/nu15102254

**Published:** 2023-05-10

**Authors:** George A. Bray

**Affiliations:** Pennington Biomedical Research Center, Louisiana State University, Baton Rouge, LA 70808, USA; george.bray@pbrc.edu

**Keywords:** BMI, body mass index, heterogeneity, food intake, energy expenditure, phenotypes

## Abstract

This review examined the origins of the concept of the BMI in the work of Quetelet in the 19th century and its subsequent adoption and use in tracking the course of the pandemic of obesity during the 20th century. In this respect, it has provided a valuable international epidemiological tool that should be retained. However, as noted in this review, the BMI is deficient in at least three ways. First, it does not measure body fat distribution, which is probably a more important guide to the risk of excess adiposity than the BMI itself. Second, it is not a very good measure of body fat, and thus its application to the diagnosis of obesity or excess adiposity in the individual patient is limited. Finally, the BMI does not provide any insights into the heterogeneity of obesity or its genetic, metabolic, physiological or psychological origins. Some of these mechanisms are traced in this review.

## 1. Introduction

“That which we call a rose By any other name would smell as sweet”; Romeo and Juliet by Wm Shakespeare. My attention to the issue of the body mass index or BMI was raised by three articles. One, a commentary by Sylvia Gonsahn-Bollie, MD, which began with the eye-catching statement “BMI is trash. Full stop.” [1]. The second was a Virtual Video conference held at the National Institutes of Health on “Moving beyond BMI: Exploring the Heterogeneity of Obesity” on 5 May 2022, which included talks by Philipp E. Scherer, Ph.D., Samuel Klein, M.D., Yvonne Commodore-Mensah, Ph.D., RN, Ruth Loos, Ph.D., Jerica M. Berge, Ph.D. and Allan Tate, Ph.D. [2]. The third was a report using various blood measurements and protein metabolites along with clinical data to develop models to predict the BMI [3]. These items suggested that a discussion of the origin, current use and future use of the BMI was worth exploring and is the purpose of this paper.

## 2. What Body Mass Index Does: Historical Perspective

Obesity is an excess of body fat or body adiposity. Evidence of obesity has been found in art work depicting the human race that can be identified in the paleolithic age, 35,000 years ago [4,5], and has been treated by all medical traditions [6]. The identification of people with obesity can be performed by simply looking at them if the degree of adiposity is sufficiently great. Many such individuals have been described in medical treatises [7]. When the degree of weight is not so great, alternative approaches may be needed to decide if someone is overweight or obese.

Weight is well known to vary with height, but the best way to express this relationship has been debated for years. One approach was proposed by the 19th century Belgian polymath, statistician and astronomer, Lambert Adolphe Francois Quetelet (1796–1874) who validated a mathematical way to estimate human body size independent of height, which could be called the “Quetelet Index”, although it is more commonly referred to as the body mass index, which is calculated as body weight (kg) divided by the square of height (m^2^) wt/kg^2^ [8,9].

Other relationships of height and weight include Weight/Height, Height/[Weight]^1/3^ (cube root of weight divided by height also called the Rorer, Corpulence or the Ponderal Index), and the weight as a percent of the average weight for that height and age in relation to body fat measured from body density and the sum of skinfolds [10]. In a seminal study, Keys and colleagues compared several of these indices to body fat estimated from densitometry as well as the sum of 2 skinfolds in 12 groups of men. His paper concludes that “Judged by the criteria of correlation with height and as a measure of body fatness the ratio of weight to height squared, or the body mass index, is slightly better than the simple ratio of weight to height, and considerably better than the ponderal index which is the poorest of the relative weight indices studied [10]. The body mass index has served as a highly useful estimate of weight relative to height for many years.

The American life insurance companies began to sound the alarm about the detrimental relationship between being overweight and early death, usually from heart disease, even before the beginning of the 20th century. Their data showed a gradual increase in the risk of death with increasing degrees of overweight [11,12].

To provide guidance to the public on the levels of desirable or ideal body weight, the Metropolitan Life Insurance company published a table of “Ideal Weight” in 1942, one for men and one for women arrayed by height according to frame size, which was divided into small frame, medium frame and large framed, without providing clear definitions for defining “frame size”. At the time I began my research in 1959, a new version of the Metropolitan Life Insurance Table called “Desirable Weight” was published and became widely used. It required height to be measured in shoes with a one-inch heel and clothing that was assumed to weigh 3 pounds. The average weight of the American population, like everything else, was gradually increasing, and to take this increase into account, the Metropolitan Life Insurance tables were updated in 1983. Because weights for height were higher than in previous tables, there was considerable concern in the health community. Other tables followed and are summarized by Kuczmarski and Flegal in their Table 2 in this reference) [13]. When the classic Framingham Study began in 1948 in the town of Framingham, MA, to study the development of cardiovascular disease, the investigators selected “Metropolitan Relative Weight” as their criterion for weight status. Relative weight has a very high correlation with the BMI (R^2^ = 0.992–0.999), essentially making them interchangeable [14]. In contrast, the correlation with body fat is considerably lower in one study (R^2^—0.68) than in others [15].

In examining the life insurance tables, I found that the lowest weight for a small-framed individual had a BMI of approximately 19 kg/m^2^ at each height. For the highest weight for a large-framed individual, the BMI was close to 24 kg/m^2^. It was an easy step to the suggestion that a BMI range of 19–24 kg/m^2^ would be a reasonable one. John Garrow in the UK suggested that a range from 20 to 25 kg/m^2^ would be easier to remember. Thus, the normal range of the BMI was set at 20–25 by the obesity community [16,17,18]. It was, however, several years before the National Center for Health Statistics switched from BMI values calculated for the top 15% of weights to a fixed value of 20–25 kg/m^2^.

A “normal range” from 20 to 25 kg/m^2^ was buttressed by a report in The Lancet, a prestigious British medical journal, showing that among 900,000 individuals, the lowest mortality occurred in the BMI range 22–24 kg/m^2^ [19]. With this impetus, the BMI became entrenched as a measurement of obesity and was used for tracking changes in the population. In this context, the BMI provided a valuable epidemiological tool that delivered the data for recognition in 1992 that the American population was becoming obese, as they continued to do for years to come [19,20,21,22].

The value of the BMI for tracking the current epidemic of obesity is clearly illustrated in the study by Rodgers et al., which traced the change in the BMI for many subgroups of the US population from 1962 to the year 2000 [23]. (See Figure 1) They showed that the US epidemic of obesity began about 1975 in all age, sex and ethnic groups and continued over the next 25 years. This fact limits the plausible explanations for the current epidemic of obesity. Rodgers and colleagues believe that it is implausible that each age, sex and ethnic group, with massive differences in life experience and attitudes, had a simultaneous decline in willpower related to healthy nutrition or exercise, or that intrauterine exposures played a major causative role. Likewise, changes in genetic make-up are unlikely to have occurred over this short period and to have affected all age groups simultaneously. Similarly, they note that it is unlikely that any factor with a long induction period had a major role in the US epidemic. Rather, they believe that the epidemic must have been caused by factors that led to rapid population-wide changes such as changes in the food supply, and I tend to agree with their conclusion.

When the BMI is sufficiently high, it is almost certainly associated with excess body fat. As Flemyng said in describing obesity some 250 years ago:

“Corpulency, when in an extraordinary degree, may be reckoned a disease, as it is some measure obstructs the free exercise of the animal functions, and has a tendency to shorten life, by paving the way to dangerous distempers”.[24]

### A High Level of the BMI Describes This Setting Clearly

The BMI has also served as the basis for estimating genetic contributions to obesity. According to Loos [2], there have been over 1700 genes related to an increased BMI. Because genes contribute approximately 40–70 percent and environment 30–60 percent, these genome-wide association studies (GWAS) can explain less than 15 percent of the variation in BMI levels.

Expanding on these GWAS studies, scientists at the University of Washington explored the relationship of multiple chemical, genetic and clinical markers as predictors of BMI [3]. As these authors note “BMI is an easily calculated and commonly understood measure among researchers, clinicians and the general public…”. BMI has a high specificity but low sensitivity in identifying individuals with excess body fat [25]. Its use as a tool for the primary diagnosis of obesity is thus limited. These authors conclude that the “BMI is unequivocally useful at the population level but too crude to capture a variety of heterogeneous metabolic health states” [3]. The table below summarizes some of the uses and limitations of the BMI.

## 3. Limitations of the BMI: What Body Mass Index (BMI) Does Not Do

Despite its value as a tool for tracking trends of weight in the population and identifying significant potential health risks in people with high BMIs, the BMI has serious limitations when it comes to evaluating individuals who may have an excess of body adiposity and for understanding the heterogeneity of obesity. Table 1 provides the author’s estimate of the value of uses of the BMI.

Several clinical guidelines for evaluating a patient with obesity suggest using the BMI as a starting point [26,27]. If the BMI is within normal limits and there is no reason to suspect lipodystrophy or sarcopenia, then the baseline BMI may make an appropriate decision point as to whether to pursue evaluation further. If the BMI is above or below the appropriate ethnic guidelines, then there are a number of steps needed to evaluate the importance of that deviation [26,27]. When the day comes that we can evaluate body fat distribution easily and inexpensively, our algorithm for approaching the patient with obesity will undergo a major change for the better.

## 4. Anatomic Limitations

The first limitation of the BMI is that it does not provide any indication of how body fat is distributed. The importance of body fat distribution was noted by the life insurance industry in the early 20th century [28,29]. Among individuals whose abdominal girth exceeded that of their chest when expanded, there was significant increased risk of mortality among insured lives. The clinical importance of fat distribution and disease received a major impetus through the seminal work of Professor Jean Vague who, in the 1940s, showed that people with central adiposity had greater health risks for cardiometabolic disease and cancer [30,31]. When comparing BMI and waist circumference as markers of cardiometabolic risk, Janssen et al. found that “…obesity-related health risk is explained by waist circumference (WC) and not by BMI. Thus, for a given WC value, overweight and obese persons have a health risk that is comparable with that of normal-weight persons.” These authors also say that “When WC and BMI were used as continuous variables in the same regression model, WC alone was a significant predictor of comorbidity” [32]. The concept that markers of adiposity were more important than weight-related ones was re-emphasized again recently [33]. Although waist circumference itself is preferable to BMI, the question of whether some other index might be better remains unclear. Several indices have been evaluated including waist circumference divided by hip circumference (WHR), weight divided by height (Wt/Height) [34,35] or WC/height^0.5^. The predictive power differs for various markers comprising the metabolic syndrome, leading this author to conclude that, at present, waist circumference is the measure of choice.

A second limitation of the BMI is that it has only a fair correlation with body fat [15,34]. A study by Gallagher et al. reported that body mass index alone accounted for only 25% of the variance in body fat in men and in women [15]. Since obesity is an excess of body fat or adiposity, we would prefer measures of fatness that are as close to the actual amount of fat as it is possible to obtain and that can be obtained at a reasonable cost. Body fat can be assessed by many methods. Anthropometric methods are among the oldest, but newer techniques greatly refine the data obtained. Bioelectric impedance and body density measurements can provide a two-compartment model with fat and non-fat tissue. Dual X-ray absorptiometry (DXA) can measure three compartments including fat, lean and bone, as well as regional location of fat by arms, legs or trunk [35,36]. Ultrasound elastography can outline tissue disease in the liver and kidney. More precise measurements come from magnetic resonance imaging (MRI) and computed tomography (CT), which can provide detailed pictures of body fat. Depending on the needs, all of these methods improve on the BMI. Current external scanning methods may provide part of the answer to obtaining better pictures of fat distribution [37,38].

A third limitation of the BMI is that it tells us nothing about the genetic, metabolic, physiological or psychological factors involved in the development of obesity. This means that in addition to expanding the anatomic basis for classifying obesity from height and weight measurements and regional fat distribution, we need information about other aspects of obesity where the BMI is silent.

## 5. Etiological Heterogeneity of Obesity

BMI is a measure of height/weight relationships and tells us nothing about the underlying heterogeneity of obesity. The earliest reports that provided specific causes for obesity came at the turn of the 20th century in reports by Babinski [39] in France and Frohlich [40] in Germany that disease at the base of the brain—the hypothalamus—could damage appetite control and produce obesity [41,42]. This was followed 12 years later by a report by the American neurosurgeon, Harvey Cushing, that a tumor of the pituitary gland could also produce obesity [43,44,45]. These clinical insights initiated a decades-long search to understand the function of the hypothalamic region of the brain in the control of food intake [46] The first theory was a dual center hypothesis proposed by Anand and Brobeck who postulated a feeding center in the ventromedial hypothalamus and an inhibitory center for feeding in the lateral hypothalamus [47]. With time and the identification of both monoamine and neuropeptide transmitters involved in feeding, the focus shifted to the arcuate nucleus with two peptides, neuropeptide Y and agouti-related peptide, stimulating feeding, and two others, pro-opiomelanocortin (POMC) and cocaine–amphetamine-related transcript (CART), inhibiting it [48]. Release of these peptides was in turn modulated by leptin, a peptide primarily produced in adipose tissue whose discovery by Friedman and his colleagues in 1994 [49] solved the mystery of why the obese mouse was obese—it lacked leptin.

## 6. Nutrients as Modulators of Obesity

The development of obesity results from a sustained positive energy balance where energy intake exceeds energy expenditure [50]. This energy comes in the form of nutrients such as glucose, amino acids, fatty acids and alcohol. The role of each of these macronutrients in the development of obesity has been an ongoing discussion. The relation of a positive energy balance is clear and the idea of whether a “calorie is a calorie” has been discussed over and over again [50].

A “glucostatic hypothesis” based on the fact that plasma glucose is highly regulated by pancreatic insulin was one of the early “nutrient specific” hypotheses designed to explain obesity. Mayer formulated a “glucostatic” hypothesis suggesting that glucose-sensing mechanisms within and outside the brain played a role in the regulation of food intake [51,52,53,54]. The role of glucose as a central nutrient continues to this day in a slightly different form as the insulin–glucose hypothesis [55].

The “lipostatic hypothesis” proposed by Kennedy suggested that fat stores were regulated [56]. This focus on body fat could be viewed as a precursor to the discovery of leptin and the ideas proposed by JP Flat that fat balance was the central regulatory mechanism [57].

Amino acids provided a third macronutrient whose dysregulation might provide a model for the development of obesity. Mellinkoff was the first to suggest this [58]. This hypothesis is also supported by the fact that the amino acid tryptophan is the precursor for the neurotransmitter serotonin, which modulates feeding [59,60].

Excess energy intake is one way to produce a positive energy flow and increased adiposity. The alternative is to reduce energy expenditure with energy intake failing to be appropriately adjusted. In an analysis of components of energy expenditure, Church et al. suggested that the decline in energy expenditure in recent decades might be enough to account for most of the positive energy balance needed to produce the levels of obesity that we see [61].

Whether the problem is excess energy intake, reduced energy expenditure or a combination, there must be some “permanent” change that occurs in the regulatory systems in the brain that prevents obesity from being easily reversed with a return to a lower weight. This contrasts with the ease with which weight is lost following overfeeding studies [62,63]. One of the most intriguing features of bariatric surgery is that the offspring tend not to be fat and develop fewer metabolic abnormalities than the mother. Metabolic/bariatric surgery or the changed metabolic milieu produced by this surgery appears to change epigenetic signals, which permits reversal of the higher set point that seems to occur with slow weight gain, making it difficult to maintain a lower weight after losing weight [64,65].

## 7. Phenotypes of Obesity

### 7.1. Metabolically Healthy Obesity

The idea that individuals can be obese and metabolically healthy and that others can be normal weight but be metabolically unhealthy has circulated for a number of years [66,67,68,69,70]. Depending on the presence or absence of increased body fat or metabolic abnormalities, four phenotypes can be defined including: (1) individuals with normal weight who are obese (NWO); (2) those that are metabolically obese but of normal weight (MONW); (3) individuals with metabolically healthy obesity (MHO); and (4) people with metabolically unhealthy obesity (MUO) who have or are at risk for metabolic syndrome [71].

There are two problems with this classification. First, the individual is identified at a single point in time. The words of the World Obesity Association indicating that “obesity is a chronic, relapsing disease process” capture this problem with the word “process” [72]. As time passes, the effects of excess adiposity may continue to afflict the individual and convert them from people who have no metabolic manifestations at the initial examination to someone with cardiometabolic consequences. It is now clear that many, if not most, individuals classified with “metabolically healthy obesity” will eventually convert to individuals with one or other disease associated with obesity [73].

The second issue confronting the use of metabolic phenotypes is related to the need for data on body fat and its distribution at different ages, taking gender into consideration. We know that women have more fat than men at the same height, age and weight. We also know that as people grow older the proportion of fat in both males and females increases, with many older people developing “sarcopenic” obesity reflected in a loss of muscle mass relative to fat mass. The relevant databases currently do not exist. The challenge of obtaining detailed information about body fat and its distribution may be solved by the use of optical scanning techniques now under development [74].

### 7.2. Phenotypes of Adipose Tissue

A number of phenotypes related to adiposity have been defined. Lipodystrophies are a rare condition, either acquired or a congenital disease, in which adipose tissue is lost, producing an angular appearance where it had previously been filled in with fat between muscle. Patients with lipodystrophy have severe metabolic disturbances, and in some of them, treatment with leptin produces dramatic responses [75]. In addition, there are lipomas that are encapsulated collections of adipose tissue [76]. Brown adipose tissue (BAT) is a third type of fat that is named because of its brownish color due to the high density of mitochondria [77]. It expresses a protein that can “uncouple” oxidation from phosphorylation and thus lead to heat production without the generation of ATP. BAT is abundant in smaller mammals and in humans at birth where it serves to produce heat to keep mammals and babies warm [78]. A second thermogenic adipocyte—the beige fat cell—can be recruited in appropriate cells in the white adipose tissue and may have similar metabolic functions once activated [79].

The predominant form of adipose tissue is composed of white adipocytes, cells whose primary function is the storage and release of fatty acids but that provide and respond to many hormonal messages. Fat cells differ in size and in their number. The concept of hyperplastic and hypertrophic forms of human obesity arose from the work of Hirsch and colleagues in New York [80] and Bjorntorp and his associates in Sweden [81]. In all forms of obesity, enlarged fat cells are a characteristic feature, but among youngsters who are obese, the number of fat cells is often significantly increased. With the enlargement of fat cells, there is often insulin resistance and an increased accumulation of fat in such places as visceral adipose tissue and the liver [82]. This regional difference in fat distribution is one of the features not captured by the BMI and one that plays a significant role in the pathogenesis of the pathology associated with obesity. It is for this reason that the American Association of Clinical Endocrinologists (AACE) labels obesity as an Adipocyte-Based Chronic Disease (ABCD) [83].

### 7.3. Genetic Phenotypes and Individual Variability

The year 1953 was a seminal one for science—the double helical structure of DNA was identified and published by Watson and Crick [84]. In the field of obesity, it led to the cloning of leptin, the first gene whose absence causes obesity [49] and was shortly identified in human families [85]. Prior to these seminal studies, familial transmission of obesity had been identified by Davenport in 1924 [86]. Studies of identical twins beginning in the 1930s [87] showed the high degree of the inheritance of obesity, even when the twins were reared apart [88]. Over the next few years, several other highly penetrant genes producing massive obesity were identified [88,89], providing clear-cut evidence of the strong genetic basis for some obesity. More than 500 genes are now known to be involved in human obesity, but only a small number, probably 15, have major effects.

Individual variability is a feature of obesity. In the overfeeding of identical twins, Bouchard and colleagues [90] found that there were significant differences in the response between pairs of male twins. This variability between individuals also manifests during weight loss including studies using diet, exercise, lifestyle, medication and surgery [91].

### 7.4. Functional Phenotypes

Functional phenotypes are another way to classify obesity. In one approach, Acosta and colleagues [92] identified four functional phenotypes, one or more of which was present in 85% of their patients. These phenotypes included: (1) “the hungry brain”, which they defined functionally as abnormal satiation; (2) emotional hunger or hedonic eating; (3) the “hungry gut”, which was defined by abnormal satiety; (4) and finally “slow burn”, which was identified as a decrease in metabolic rate. In 15% of participants, no phenotype was identified, and 27% of their patients had two or more phenotypes. The fact that so many people did not fit into their categories suggests that other phenotypes must exist or that their phenotypes need to be refined.

## 8. Conclusions

The importance of the BMI in replacing height and weight tables cannot be denied. A high value of the BMI is almost certainly a good guide to the need for additional evaluation for potential metabolic abnormalities and complicating diseases. Diagnosing potential risks from obesity using the range of BMI values might be analogous to the range of BP in diagnosing the risk of stroke, cholesterol in diagnosing the risk of myocardial infarction or smoking in diagnosing lung cancer [93]. They each indicate increased risk but do not make a “diagnosis”. However, as a tool for tracking population changes, we do not presently have a better tool.

On the other hand, for evaluating the individual patient, the BMI leaves much to be desired and requires additional information, specifically about fat distribution and about the impacts that the increased adiposity is having on the individual. When the BMI or other clinical markers leave the healthcare provider with concerns, he/she should measure central adiposity with waist circumference at a minimum and use other appropriate imaging techniques such as DXA or scans.

## Figures and Tables

**Figure 1 nutrients-15-02254-f001:**
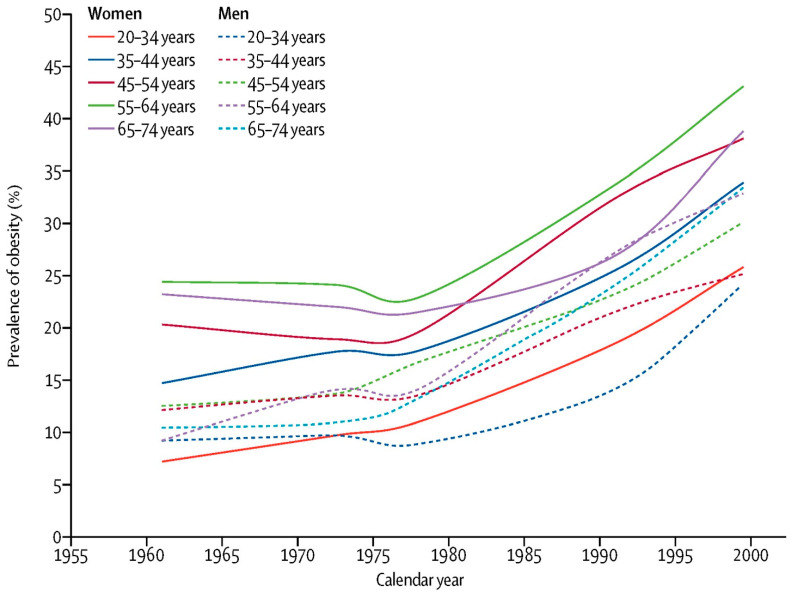
Prevalence of obesity estimated by the body mass index from 1962 to 2000.

**Table 1 nutrients-15-02254-t001:** Uses and limitations of the BMI.

Measurement	Value Rating
Estimation of body weight	****
Tracking population weight	****
Estimation of body fat	**
Estimating distribution of fat	0
Use in genetic studies	**
Pathophysiology of obesity	0
Phenotyping obesity	0

0 = not useful information; ** = some useful information; **** = valuable.

## Data Availability

All data comes from published ources shown in the references.

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
