# Peer review of "Beyond BMI"

_nutrients, 2023, doi:10.3390/nu15102254_

Round 1

Reviewer 1 Report

After reading this nice review of the value and limits of using BMI, I was hoping for some practical recommendations at the end for the practicing clinician.  Instead the author ends with stating the problem and need for more work in the future.

Would recommend the following content be added:

1. Importance of routine measurement of the waist circumference as a component of the metabolic syndrome

2. Value of body composition analysis available with DXA machines

3. Value of assessing liver fat with transient elastography

Would then add to the conclusion that in patients where there is a question if an elevated BMI represents excess adipose tissue or muscle, getting a DXA scan or elastography study can be helpful.

few minor errors or typos

Author Response

Reviewer 1 for Beyond BMT 2023.05.01

After reading this nice review of the value and limits of using BMI, I was hoping for some practical recommendations at the end for the practicing clinician.  Instead the author ends with stating the problem and need for more work in the future.

RESPONSE:  I have modified the Conclusion to note the importance of assessing central adiposity and measuring appropriate markers of the metabolic syndrome

Would recommend the following content be added: 

RESPONSE: Thank you for the suggestions.  I have tried to comply and added a few new references.

  1. Importance of routine measurement of the waist circumference as a component of the metabolic syndrome

RESPONSE  P 7 I have added the material below to broaden the discussion

“The concept that markers of adiposity were more important than weight related ones was re-emphasized again recently (Criminisi A, Sorek N, Heymsfield SB. Normalized sensitivity of multi-dimensional body composition biomarkers for risk change prediction.   Sci Rep. 2022 Jul 20;12(1):12375. doi: 10.1038/s41598-022-16142-1. PMID: 35858946)

Although waist circumference itself is preferable to BMI, the question of whether some other index might be better remains unclear.   Several indices have been evaluated including waist circumference divided by hip circumference (WHR); weight divided by height (Wt/Height) (Ashwell M, Gibson S.  Waist-to-height ratio as an indicator of 'early health risk': simpler and more predictive than using a 'matrix' based on BMI and waist circumference.   BMJ Open. 2016 Mar 14;6(3):e010159. doi: 10.1136/bmjopen-2015-010159. PMID: 26975935) or the WC/height0.5 (Nevill AM, Duncan MJ, Myers T.   BMI is dead; long live waist-circumference indices: But which index should we choose to predict cardio-metabolic risk?    Nutr Metab Cardiovasc Dis. 2022 Jul;32(7):1642-1650. doi: 10.1016/j.numecd.2022.04.003. Epub 2022 Apr 10. PMID: 35525679) have been evaluated.  The predictive power differs for various markers comprising the metabolic syndrome leading this author to conclude that at present waist circumference is the measure of choice.     

  1. Value of body composition analysis available with DXA machines

RESPONSE P 7 This was added:  I have broadened your request by talking about measurement of body composition with a focus on fat.  Here is what was added   NEW Content:   

Body fat can be assessed by many methods.  Anthropometric methods are among the oldest, but newer techniques greatly refine the data obtained.  Bioelectric impedance and body density measurements can provide a 2-compartment model with fat and non-fat tissue.  Dual x-ray absorptiometry (DXA) can measure 3 compartments including fat, lean and bone as well as regional location of fat by arms, legs or trunk.  Ultrasound elastography can outline tissue disease in the liver and kidney.  More precise measurements come from magnetic resonance imaging (MRI) and computed tomography (CT) which can provide detailed pictures of body fat.  Depending on the needs all of these methods improve on BMI]

  1. Value of assessing liver fat with transient elastography. I have included it, but my reading of the methodologic paper by Ozturk et al from the MGH had little to do with fat. I have thus mentioned it but little else. 

RESPONSE

Would then add to the conclusion that in patients where there is a question if an elevated BMI represents excess adipose tissue or muscle, getting a DXA scan or elastography study can be helpful.

RESPONSE:  Done as requested

Comments on the Quality of English Language few minor errors or typos

Reviewer 2 Report

I have been reading this paper carefully and methodologically I have nothing to say, from my point of view I consider it of great interest to the scientific community, and its contribution is clear from the first moment. If all the articles I review were like this, it would be a pleasure for the eyes and for the readers of the Journal.

Author Response

Thank you for the very nice comments.  Wish all reviewers were like you!